# Factors Affecting Pollination and Pollinators in Oil Palm Plantations: A Review with an Emphasis on the *Elaeidobius kamerunicus* Weevil (Coleoptera: Curculionidae)

**DOI:** 10.3390/insects14050454

**Published:** 2023-05-11

**Authors:** Christharina S. Gintoron, Muhamad Azmi Mohammed, Siti Nurlydia Sazali, Elvy Quatrin Deka, Kian Huat Ong, Imran Haider Shamsi, Patricia Jie Hung King

**Affiliations:** 1Faculty of Agricultural and Forestry Sciences, Universiti Putra Malaysia Bintulu Sarawak, Jalan Nyabau, Bintulu 97008, Sarawak, Malaysia; 2Institute Ecosystem Science Borneo, Universiti Putra Malaysia Bintulu Sarawak, Jalan Nyabau, Bintulu 97008, Sarawak, Malaysia; 3Center for Pre-University Studies, Universiti Malaysia Sarawak, Kota Samarahan 94300, Sarawak, Malaysia; 4Faculty of Resource Science and Technology, Universiti Malaysia Sarawak, Kota Samarahan 94300, Sarawak, Malaysia; 5Key Laboratory of Crop Germplasm Resource, Department of Agronomy, College of Agriculture and Biotechnology, Zhejiang University, Hangzhou 310058, China

**Keywords:** *Elaeidobius kamerunicus*, oil palm, pollination, weevils

## Abstract

**Simple Summary:**

Pollination plays a major role in oil palm yield. Oil palm inflorescences emit an anise-like scent that draws pollinators to alternate between male and female inflorescences, transferring the dusted pollens at once, which leads to fruit formation. Forthright as it may sound, however, its efficiency is determined multifactorially. This review examines both abiotic and biotic factors that may influence pollination and pollinators in oil palm plantations, with particular attention given to the pollinating weevil, *Elaeidobius kamerunicus*. As primary pollinators that rely on oil palm male inflorescences for food and breeding, the unique traits that contribute to their success are also explored. However, their population size and pollinating efficiency can be influenced by various factors, such as weather, palm varieties, soil types, and pollen quality and quantity. Other factors that can impact their success include the presence of parasitic nematodes and predators, insecticide application, weevil endogamy, and the proximity of palms to natural forests. Finally, research gaps and recommendations are discussed.

**Abstract:**

Pollination is crucial for oil palm yield, and its efficiency is influenced by multiple factors, including the effectiveness of *Elaeidobius kamerunicus* weevils as pollinators in Southeast Asia. Weevils transfer pollen between male and female flowers, leading to successful fertilization and fruit development, which contributes to higher oil palm yields and increased production of valuable oil. Understanding and conserving the weevil population is important for sustainable oil palm cultivation practices. The interaction between pollinators, including weevils, and environmental factors is complex, involving aspects such as pollinator behavior, abundance, diversity, and effectiveness, which are influenced by weather, landscape composition, and pesticide use. Understanding these interactions is critical for promoting sustainable pollination practices, including effective pest management and maintaining optimal pollinator populations. This review discusses various abiotic and biotic factors that affect pollination and pollinators in oil palm plantations, with a particular focus on weevils as primary pollinators. Factors such as rainfall, humidity, oil palm species, temperature, endogamy, parasitic nematodes, insecticides, predators, and proximity to natural forests can impact the weevil population. Further research is recommended to fill knowledge gaps and promote sustainable pollination practices in the oil palm industry.

## 1. Introduction

According to the Food and Agriculture Organization [1], pollination plays a crucial role in the production of both food and non-food crops, with about 75% of global food crop species relying at least partially on pollination. This makes pollination economically important for agriculture and global food security, as it affects crops, such as cotton, and biofuel crops as well. However, pollinator populations are declining globally due to multiple factors such as habitat loss, pesticide exposure, disease, climate change, and changes in land use practices, which has significant impacts on crop production. Reduced pollination services can lead to decreased crop yields, lower quality fruits and seeds, and potential cascading effects on ecosystem functioning and biodiversity.

In recent years, crop pollination has become endangered [1,2,3], with declines in pollinator populations and their ability to effectively pollinate crops due to a combination of several factors including habitat loss caused by human activities, such as agriculture and urbanization, which removes specific habitats for nesting and foraging. The use of pesticides, particularly neonicotinoid insecticides, has also been linked to pollinator population declines by affecting their behavior, navigation, and reproductive success. Climate change, including changes in temperature, rainfall patterns, and extreme weather events, can also disrupt the timing of pollinator emergence and migration, as well as the availability of food, leading to further declines in populations. The introduction and spread of non-native pathogens and parasites can also have devastating effects on pollinator populations. Finally, inadequate pollen quality or quantity can also contribute to poor crop pollination, leading to lower yields [4].

The oil palm ecosystem is particularly vulnerable to the negative impacts of these factors on pollinators and crop pollination. This is because oil palm trees require cross-pollination to produce fruit, and the extent of cross-pollination can affect the yield of the crop. Palm oil-producing countries including Sierra Leone, Nigeria, Colombia, Thailand, Malaysia, and Indonesia are grappling with a decline in oil palm fruit set (FS), leading to a decrease in fresh fruit bunch (FFB) and oil yield [5]. This decline in fruit set is particularly worrisome for oil palm growers in Malaysia, where reports of low fruit set have resulted in a significant reduction in fresh fruit bunch yield, from 17.16 tons per hectare (t/ha) in 2018 to 17.89 t/ha in 2017 [6]. Therefore, in the following discussion, we will focus specifically on the oil palm ecosystem and examine how each of these factors affects pollinators in this system, as well as their potential impact on crop pollination. By understanding these factors and their effects on pollinators in the oil palm ecosystem, we can better develop strategies to maintain the pollinator populations and ensure effective crop pollination in this important agricultural system.

Oil palm (*Elaeis guineensis* Jacquin) is a monoecious plant, with separate male and female inflorescences on the same tree. The female inflorescence of oil palm is crucial for the production of fruit. It is from the female inflorescence that the fruit bunches develop, and ultimately the oil that is extracted from the fruit. If the female inflorescence is not pollinated, the fruit will not develop, resulting in reduced yields or even complete crop failure. Therefore, effective cross-pollination is essential for oil palm cultivation and the productivity of the crop [7]. The female inflorescences are found on the axillary branches and form fruit bunches. Each bunch may contain up to several hundred individual fruits. In contrast, the male flowers are located on the spike-like inflorescences that emerge from the axils of the leaves. The male inflorescence comprises a central axis that bears many branches, each containing numerous small flowers. The male flowers produce pollen, which is then transferred to the female flowers by the wind (anemophil pollination) and various insects, such as weevils (entomophil pollination) [8].

Fruit set formation in oil palm is closely related to pollination. The transfer of pollen from the male to the female inflorescences is necessary for fertilization and subsequent fruit development. Studies have shown that effective pollination can increase oil palm yield by up to 50% [9]. Pollination can be affected by various factors, including the abundance and diversity of pollinators, the quality and quantity of pollen, and the timing of pollination relative to flowering. Additionally, oil palm pollen has a short lifespan of less than 24 h, which further emphasizes the importance of timely and efficient pollination [10]. Various pollination techniques have been employed in Malaysia since the establishment of oil palm plantations to improve fruit set formation, with a few of these techniques summarized in Table 1.

Ref. [17] classified the main types of pollination in oil palm plantations into four categories: anemophily (pollination by wind), entomophily (pollination by insects), zoomophily (pollination by animals, such as birds and bats), and artificial pollination (pollination by humans). Oil palm trees can be pollinated by both wind and insect pollination. This is facilitated by the presence of certain floral characteristics such as plentiful smooth pollen grains, extended stigmatic surfaces, and vibrant colorful corolla [17]. Ref. [15], however, recorded that a greater fruit set is produced by weevil pollination compared to wind. Hence, entomophilous pollination is the most crucial pollination method, especially among young palms [18,19]. While various insects, such as *Thrips hawaiiensis* and *Pyroderces* sp., can pollinate *Elaeis* palm, weevils of the genus *Elaeidobius*, such as *Elaeidobius kamerunicus*, are the most effective [19,20,21].

## 2. Oil Palm Pollination Agents

During the anthesis stage of oil palm, male inflorescences attract more insect pollinators compared with females. Previous studies conducted in Malaysia, Venezuela, Kerala, and Cote d’Ivoire stated that approximately only 1% of the insect that visits male inflorescences are present in females [22,23,24,25]. Several *Elaeidobius* species in West Africa, such as *E. kamerunicus*, *E. plagiatus* (Fabricius), *E. singularis* (Faust), *E. bilineatus* (Hustache), and *E. subvittatus*, are found in both male and female inflorescences [26]. In Malaysia, *Thrips hawaiiensis* (Thysanoptera) and Pyroderces (Lepidoptera) are regarded as native oil palm pollinators [27]. However, according to reports by Lepesme and Syed as cited in [28], *Pyroderces* spp. has a strong preference for female inflorescences and is active for only two to three hours post-sunset [29], leading to the belief that these insects do not contribute to pollination services in oil palm. Conversely, *Thrips hawaiiensis*, known for its limited flying ability, especially during rainy days [30], tends to avoid open and windy areas, even when in close proximity to older palms, as noted by Lewis and Syed, as cited in [28]. The inefficiency of these insects in pollinating oil palm led to the introduction of *E. kamerunicus* in the early 1980s [5,31,32,33,34].

### 2.1. Biology of Elaeidobius kamerunicus

*Elaeidobius kamerunicus*, also known as the oil palm weevil, is native to the West African oil palm [27]. The weevils are highly selective in their feeding behavior, preferring to feed only on oil palm flowers and rejecting other plant species [35]. This feeding behavior is critical for the weevils’ survival and reproduction because it provides them with the energy and nutrients; they need to lay eggs and produce offspring.

Once the female weevils have fed on the oil palm flowers and mated with the male weevils, they lay their eggs in the flower’s bracts or the male flowers’ bases [36]. The eggs hatch into larvae, which then feed on the developing flowers of the oil palm tree [37]. The larvae have specialized mouthparts that allow them to feed on the soft tissues of the developing flowers [38]. As the larvae mature, they pupate and develop into adult weevils, completing the life cycle. *E. kamerunicus* spent most of its life cycle in the male inflorescences of oil palms [17,27,39]. Its reproduction rate is strongly associated with the availability of male inflorescences apart from feeding on the anther filaments [39,40]. Being a holometabolous insect, this weevil species develops through different stages: egg, larva, pupa, and imago [41,42].

### 2.2. Roles of VOCs in Oil Palm Pollination

*E. kamerunicus* is reported to favor a certain concentration of estragole. This trait is linked to “multiple functions” in which the weevil’s sensory function is impaired by the high concentration of the compound [43]. *E. kamerunicus* is reported to be less attracted at concentrations of estragole lower or higher than 100 ppm [43]. This confers a sustainable advantage to the oil palm as this insect’s behavior enables the insect to forage among healthier palms and avoid unhealthy palms, which often produce higher volatile organic compounds or VOCs than favored by *E. kamerunicus* [43].

However, abiotic and biotic stress could alter this behavior and induce the *E. kamerunicus* to be lured at higher concentrations of VOCs [44]. Such stresses include high temperature, high light intensity, and herbivores’ attacks. Limited information is available regarding the ecology of interactions between *E. kamerunicus* and *Elaeis guineensis* inflorescences, hindering the recommendation of optimal environmental conditions for successful pollination. However, weevil activity is influenced by inflorescence sex ratio, temperature, and rainfall. One study found significant variations in estragole emission and intensity among oil palm varieties and inflorescences, affecting interactions with pollinators, such as Curculionidae weevils [45]. Additionally, the composition of estragole differs depending on soil type, with mineral, sandy, and peat soils exhibiting the lowest to highest amounts, respectively. Estragole constitutes a significant proportion of volatile organic compounds released by palms on sandy soil (37.49%), clay soil (30.71%), and peat soil (27.79%) [35].

Male and female oil palm inflorescences emit different blends of volatile compounds, which the oil palm weevil can detect and use to navigate between the two types of flowers. Generally, male flowers are regarded as more rewarding than female flowers [46] to the pollinators as they provide pollen, nectar, and shelter in exchange for visits, whereas female flowers only offer nectar [46]. However, female oil palm inflorescences have a way to deceive the weevils by producing a similar scent to male flowers, in which both flowers produce a scent containing a certain percentage of palmitic acid. In addition to volatile compounds, visual cues may also play a role in attracting oil palm weevils to inflorescences. According to [36], oil palm weevils are attracted to the bright yellow color of the male flowers.

The movement of *E. kamerunicus* from male to female inflorescences in oil palms is likely due to the distribution of volatile compounds released by the inflorescences. The volatile compounds produced by oil palm inflorescences include (a) Estragole (C_10_H_12_O), a phenylpropene organic compound that is the major compound emitted by male oil palm flowers at all flowering stages and is responsible for attracting *E. kamerunicus* weevils to the male flowers for pollen collection [47]; (b) palmitic acid and 1-dodecyne, which are other volatile compounds produced by male inflorescences at all flowering stages [31]; (c) 9,12-Octadecadieonic acid (C_18_H_32_O_2_), n-Hexadecanoic acid (C_10_H_12_O), and dan 9,12,15-Octadecatrienoic acid (C_18_H_30_O_2_), which are other reported volatile compounds released by male inflorescences [47]; and (d) Farnesol and squalene, which are volatile compounds produced by female oil palm inflorescences at all flowering stages, and due to the synchronicity of anthesis with the male inflorescences, as well as the similar scent released, the weevils may visit them by mistake [20,31].

Both male and female oil palm inflorescences emit volatile fragrances during the flowering phase, with males producing a stronger scent [48]. Female flowers typically bloom earlier than male flowers, opening on the third day of anthesis and reaching the peak blooming time between 8:00 and 9:30 a.m. Male flowers usually open on the fourth day, reaching the peak blooming time between 9:00 and 10:00 a.m. Weevils are more attracted to male inflorescences during the first to sixth day of anthesis, while female inflorescences mostly aggregate during the first to fifth day.

Weevils are attracted to the scent emitted by female flowers, and this attraction increases when male flowers start to wilt. The emission of fragrance in female inflorescences is influenced by temperature changes during pollination [17]. When the temperature rises during the opening and anthesis of female flowers, it alters the volatilization of substances that generate a strong scent, thereby attracting weevils that previously visited male flowers. This suggests that volatile compounds play a crucial role in the weevils’ ability to locate female flowers after visiting male flowers. The concentration of estragole decreases in male inflorescences as they start to wilt, while farnesol produced by female inflorescences increases [31]. This change in concentration may also contribute to the attraction of the weevils to female flowers, resulting in the transfer of pollen from male to female flowers during the process.

### 2.3. Key Characteristics of E. kamerunicus as the Main Pollinator

*Elaeidobius kamerunicus* is the main pollinator for oil palm after its introduction to Asia in the 1980s. Their success is partly contributed to their high capacity to transfer pollen and because they are extremely host-specific (Table 2). *E. kamerunicus* is host-specific to oil palm trees (*E. guineensis*) [49]. The close relationship between *E. kamerunicus* and the oil palm inflorescences enhances the pollen transportation and development of fruit sets [42]. This mutualistic relation between weevils and oil palm is exploited by the oil palm industry, and the level of non-parthenocarpic fruit in the fruit set had increased to approximately 36% after the introduction of *E. kamerunincus* into Malaysian oil palm estates [17].

#### 2.3.1. Adapted Mouthparts

One of the factors that make *E. kamerunicus* host-specific to oil palm trees is the structure of the weevil’s mouthparts, which are adapted to fit the flower morphology of the oil palm [50]. The flowers of oil palm trees have a unique morphology that includes three stamens and three stigmas, which are located at the base of the flower [30]. The mouthparts of the weevil have developed a a long, narrow snout that can reach the base of the flower and be in contact with the stigmas and stamens. The snout is also equipped with sharp mandibles that can cut through the tough outer layer of the flower to reach the reproductive organs. Furthermore, the mouthparts of the weevil are also adapted for feeding on the pollen and nectar of the oil palm flower. The snout has a grooved surface that allows the weevil to collect pollen grains, while the mandibles are used to scrape the nectar from the flower. Additionally, the female weevils have specialized antennae that can detect the presence of specific floral odors emitted by oil palm flowers, making it easier for them to locate the flowers [51].

#### 2.3.2. Complementarity with Native Pollinators

Another key factor that attributes *E. kamerunicus* as the main pollinator of palms is its ability to coexist with other insect pollinators and complement their pollination services. In Malaysia, even in cases where the weevil population is low, the presence of other native pollinators, such as *T. hawaiiensis* and *Pyroderces* sp., have been observed to sustain fruit sets above 60%, indicating that these pollinators have not been displaced but rather complement each other in providing this crucial ecosystem service [34]. This coexistence of different pollinators is mainly contributed to their unique habitat and food requirements. Although they may compete for common food resources, they have different host plants, adaptability to rainfall, and activity times, as evidenced by studies conducted in Malaysia and Indonesia and reviewed by [31,33,34] (Table 3). For instance, *E. kamerunicus* and *Pyroderces* sp. pupate in male oil palm flowers, while *T. hawaiiensis* pupates in soil [27,34]. This means that the different pollinators are not in direct competition for resources, and each can fulfill its unique role in the pollination of oil palm flowers, ultimately leading to higher fruit sets and yields.

*E. kamerunicus* tends to forage primarily in the morning to midday, around 1000 to 1200 [20,31,52], while *T. hawaiiensis* exhibits peak activity during two different periods, between 0800 and 0900 and 1400 and1500 [20,31,52]. Additionally, *T. hawaiiensis* is better adapted to the dry season and maintains optimal pollination services during this time [34]. While climatic factors impact pollination activity, *E. kamerunicus* is less affected compared to other insect pollinators, allowing it to maintain pollination services even during the wet season [27,40].

#### 2.3.3. High Pollen Carrying Capacity

*E. kamerunicus* is recognized as one of the most efficient pollinators in the oil palm landscape, as it has the highest capacity for pollen cartage. Adult weevils of this species carry the highest average number of pollen grains [18,33,52,53,54,55]. On average, *E. kamerunicus* can carry an average of 80,000 to 100,000 pollen grains per visit to the female inflorescence of oil palm [55]. Ref. [55] observed that the amount of pollen carried by *E. kamerunicus* is positively correlated with their body size, with larger weevils carrying higher pollen loads. Bristles or hairs on the body surface also contribute to higher pollen loads, as reported by studies, such as [17,18].

Male weevils have been found to carry more pollen than females, and the pollen is primarily found on their elytra, thorax, and abdomen [55]. Male weevils have a body size ranging from 3–4 mm and possess more fine hairs or bristles on their abdomen, which aid in enhancing pollen transfer. These pleural bristles are not present in females [12,18,42,56]. This trait is a significant determinant of the pollinators’ success in performing their ecosystem service, as a higher number of pollen grains carried in each visit to the female inflorescence increases the likelihood of successful pollination [18].

#### 2.3.4. Ability to Select High-Quality Pollen

Interestingly, there have been several studies that suggest *E. kamerunicus* has the ability to select high-quality pollen during their foraging activities. For example, ref. [55] reported that *E. kamerunicus* showed a preference for fresh pollen, which is known to contain higher levels of viable pollen grains. The study also found that the weevils avoided inflorescences with stale or aged pollen. In another study, ref. [54] reported that *E. kamerunicus* showed a preference for pollen from younger male inflorescences over older ones. The younger inflorescences are known to produce higher-quality pollen, which may explain the preference of *E. kamerunicus* for them. *E. kamerunicus* also showed a preference for inflorescences with higher pollen availability [53]. Ref. [18] reported that the transported pollen grains are of very good quality, with a germination rate of more than 70%. The study suggested that this behavior may be an adaptation to optimize foraging efficiency by reducing the amount of time spent searching for pollen sources.

Overall, these previous studies suggested that *E. kamerunicus* may possess the ability to identify and preferentially select pollen sources that lead to successful pollination, potentially through a preference for a specific concentration of estragole. The variation in estragole concentration among different flower ages may serve as a signal for *E. kamerunicus* to locate high-quality pollen. *E. kamerunicus* is an effective pollinator of oil palm, and it is important to preserve and promote the natural populations of *E. kamerunicus* in oil palm plantations, as they play a critical role in the pollination and reproduction of the oil palm trees.

### 2.4. Pollinator Health

Pollinator health could be a possible factor contributing to the declining *E. kamerunicus* population size. Factors such as endogamy of weevils and genetic bottleneck, pesticide use, habitat loss, and disease can all have negative impacts on the health and population size of pollinators.

#### 2.4.1. Endogamy of Weevils and Genetic Bottleneck

*E. kamerunicus* was introduced into Asia due to its high reproductive rate and good searching ability, and primarily due to its close association with oil palm male inflorescences, being host-specific and numerous in both dry and wet weather [33,57,58,59]. Following its arrival in Malaysia between July and December 1980, *E. kamerunicus* was present in virtually all oil palm estates in Malaysia by April 1982 [58]. As there have been no further introductions since then, it is assumed that these pollinating weevils may have a limited genetic basis, leading to inbreeding depression and negatively influencing their pollinating efficiency [28].

Inbreeding depression occurs when homozygosity increases due to reduced genetic diversity, leading to the expression of deleterious recessive mutations [59,60]. Bottlenecks and reduced population size result in mating among small and closely related individuals within the introduced population [60]. The loss of progeny fitness is a major impact of inbreeding depression [59,61,62]. In a study of leaf beetles [61], inbreeding depression caused a decrease in larval and adult survival, a decline in reproductive output, and prolonged larval development, while other observations reviewed by [28] included a compromised immune system (*Formica exsecta*) and an impacted sex ratio (*Uscana semifumipennis*).

The morphological evolution of domesticated weevils has been documented by [63,64], who found a distinct separation between samples from Malaysia and Cameroon, and also among samples from Malaysia, Indonesia, and Liberia. This indicates a direct relationship between morphometric divergence and geographical separation. In 2020, the first study focusing on the genetic information of *E. kamerunicus* in Malaysia, Indonesia, and Cameroon was reported [21]. The main finding showed different genetic diversity in *E. kamerunicus* populations in Indonesia and Malaysia compared to those from Cameroon, with decreased genetic diversity observed in Malaysia and Indonesia. Population size is one factor that could cause declining genetic diversity, which is expected given that the first and only introduction was between 1980 and 1982 [21]. Additionally, genetic differentiation analysis showed genetic variation between weevil populations from these three countries [21].

Meanwhile, ref. [65] suggested that the *E. kamerunicus* population introduced to Asia and South America may have experienced a genetic bottleneck, which is supported by [28] as the introduced weevils are originally from Western and Central Africa, and the introductions were made from a small number of weevils, resulting in a limited genetic base among these populations. This is congruent with the low haplotype number across all Asian weevil populations reported by [65].

A study by [66] used microsatellite markers to investigate the genetic diversity and structure of *E. kamerunicus* populations in two oil palm plantations in Malaysia. The study found that the populations had low levels of genetic diversity and high levels of genetic differentiation, which could be attributed to factors such as isolation and founder effects. The authors suggested that the low genetic diversity and high genetic differentiation could have implications for the effectiveness of biological control methods for *E. kamerunicus*. Another study by [67] also used microsatellite markers to examine the genetic diversity and structure of *E. kamerunicus* populations in oil palm plantations in Cameroon. The study found that the populations had moderate levels of genetic diversity but high levels of genetic differentiation, which could be due to factors such as habitat fragmentation and limited gene flow.

However, some studies have presented contrasting results. For instance, a genetic diversity study of *E. kamerunicus* in Indonesia reported high levels of genetic exchange, resulting in genetic variation that sustained the survival of the species in the region [68]. Most populations in Indonesia were found not to result from inbreeding, suggesting that the introduced weevils have maintained a healthy genetic diversity. This phenomenon may be attributed to the “purging” of deleterious recessive alleles that are responsible for inbreeding depression, which can occur even in natural conditions [28,60]. This could explain the lack of inbreeding observed in the introduced *E. kamerunicus* populations in Southeast Asia [68].

Overall, while there is limited information on the endogamy and genetic bottleneck of *E. kamerunicus*, these studies suggest that the populations may have low genetic diversity and high genetic differentiation due to factors such as isolation, limited dispersal, and habitat fragmentation. Future research should focus on how genetic bottleneck affects the pollination efficiency of *E. kamerunicus* [68]. Additionally, more genetic diversity studies are needed to better understand the evolutionary implications for the population structure of these introduced weevils [63]. Inbreeding depression is an essential component that should be of concern in population management, as it bridges population genetics.

#### 2.4.2. Nematodes Parasitism

Nematodes parasitism in *E. kamerunicus* has been reported in several studies. One study found that *E. kamerunicus* individuals were heavily infected with nematodes, with an average of 6.5 nematodes per individual and a prevalence of 92% [69]. These nematodes were identified as belonging to the genus *Sphaerularia*, which is known to parasitize weevils [69,70]. In another study, a different nematode species, *Cosmocercoides* sp., was found in the gut of *E. kamerunicus* individuals, with a prevalence of 75% [71]. Nematode parasitism can have negative impacts on the host weevil. In one study, the presence of *Sphaerularia* nematodes in *E. kamerunicus* caused a reduction in the weevils’ body size, reproductive output, and survival [69]. Additionally, nematodes can also cause physical damage to the host’s gut tissue, which could lead to impaired digestion and other health issues [72].

It is not entirely clear how nematodes are transmitted to *E. kamerunicus*, but it is possible that they are acquired through feeding on infected plant material or from contact with other infected weevils [71]. Control of nematode parasitism in *E. kamerunicus* is challenging, as there are no effective treatments or management strategies currently available. However, reducing stress on the weevil population by sustainable agricultural practices and minimizing disturbances to their habitat may help to reduce the impact of nematode parasitism.

In Malaysia, it has been reported that poor fruit formation (10–20%) is associated with parasitic nematode infection [73]. Entomopathogenic nematodes are considered deadly obligatory parasites that inhabit the hemocoel and rely on their hosts [72,74]. These nematodes have been found to reduce the longevity and fecundity of weevils by decreasing egg production [74,75]. The depletion of fat reserves in infected weevils leads to smaller size and sterility, which may potentially affect the pollination efficiency of *E. kamerunicus* [74,75].

#### 2.4.3. Predatory Threats

Predators, such as insectivorous rodents, birds, insects in the orders of Heteroptera, Hymenoptera, and Diptera, and spiders, can significantly influence the population dynamics of *E. kamerunicus* and affect their pollination behavior [13,52,76,77]. Omnivorous rodents, such as *Rattus tiomanicus* and *Rattus tanezumi*, feed on a variety of food sources, including *E. kamerunicus*, and can reduce weevil populations through predation. According to Liau et al.’s findings, as cited in [78], *Rattus tiomanicus* feeds on and causes significant damage to oil palm inflorescences. When rats graze on male spikelets, they are likely to consume the eggs, larvae, and pupae that reside inside them. Birds, such as *Parus major*, the *Prinia/Orthotomus* group, and yellow-vented bulbul *Pycnonotus goiavier* (Pycnonotidae), also play a vital role in the predation of *E. kamerunicus* in oil palm plantations. Some Sylviidae bird species, such as ashy tailorbird *O. ruficeps* and black-throated *P. atrogularis*, have even modified their feeding behavior to target *E. kamerunicus* exclusively [79], potentially reducing the weevil population and affecting oil palm fruit-setting during early production years. For example, research has shown that black-throated *Prinia* in Indonesian oil palm ecosystems primarily feed on *E. kamerunicus*, which constitutes more than 75% of their diet. These birds specifically target male inflorescences during anthesis, making them vulnerable, as reported by [76]. However, the impact of these predators on *E. kamerunicus* populations and their pollination activity is not fully understood, and more research is needed to clarify their significance [27].

Additionally, various insects in the orders of Heteroptera, Hymenoptera, and Diptera serve as predators of *E. kamerunicus*. Heteropterans, such as assassin bugs *Cosmolestes picticeps* (Reduviidae) and Sycanus *dichotomus* (Reduviidae), prey on adult *E. kamerunicus* [80]. They insert their elongated pointed proboscis into the body of the *E. kamerunicus* and suck out its bodily fluids. These insects were originally introduced into oil palm landscapes to control the population of bagworm *Metisa plana* (Lepidoptera) in oil palm plantations, as reported by [81]. However, when the population of *M. plana* is low, *E. kamerunicus* becomes vulnerable and easily targeted.

Hymenopterans, including certain wasps, big-headed ants, and *Pheidole megacephala*, prey on adult *E. kamerunicus* by using their large and powerful mandibles to tear open the outer part of the body [78]. Dipterans, such as hoverflies and robber flies, are also known to prey on *E. kamerunicus* in their larval or adult stages. Furthermore, spiders, which are abundant in many ecosystems, can capture *E. kamerunicus* in their webs or actively hunt them. Frequently, adult *E. kamerunicus* are discovered ensnared in the webs of web-building spiders, such as *Argiope* sp. (Araneidae) and *Leucauge grata* (Tetragnathidae) [78]. The *Argiope* sp. typically constructs its web on ferns (*Lygodium flexuosum*) growing on the trunk of oil palm trees, while *L. grata* builds its web in open spaces approximately one to two meters above ground in oil palm ecosystems. Overall, these predators can significantly influence the population dynamics of *E. kamerunicus* by preying on them at different life stages. Understanding the role of these predators in the ecosystem is crucial for the conservation and management of E. kamerunicus populations and for maintaining the balance of the overall ecosystem.

#### 2.4.4. Use of Pesticides

Pesticide use, particularly the use of neonicotinoid insecticides, has been linked to declines in pollinator populations. These insecticides are toxic to bees and other pollinators and can cause disorientation, impaired learning and memory, and reduced reproductive success [80]. Insecticide application is a common practice in oil palm plantations, but it can be harmful to *E. kamerunicus* [13], which is a primary pollinator of oil palm [5,12]. The use of insecticides is a major contributing factor to changes in insect populations, and the high susceptibility of *E. kamerunicus* to cypermethrin and chlorantraniliprole, two commonly used insecticides, is a significant concern [53,76]. However, ref. [82] reported that *E. kamerunicus* is more susceptible to cypermethrin compared to chlorantraniliprole. The study found that cypermethrin caused higher mortality in *E. kamerunicus* compared to chlorantraniliprole.

Nevertheless, it is important to note that different studies may report different results based on factors such as the concentration of insecticide used, exposure duration, and environmental conditions. Therefore, it is crucial to consider the specific context of each study when interpreting the results. Most studies agree that female weevils are less susceptible to insecticides than males due to higher detoxification activity, and larger individuals generally have higher tolerance [83,84]. This is likely due to higher detoxification activity in females. Ref. [84] found that male *E. kamerunicus* were more susceptible to chlorpyrifos and cypermethrin insecticides than females. To ensure sustainable oil palm production and the conservation of pollinator populations, it is crucial to consider the biology of the targeted species and minimize harmful effects on beneficial insects when implementing chemical treatments in plantations.

## 3. Inadequate Pollen Deposition and Pollen Quality

In oil palm plantations, pollen quality and quantity are essential factors in crop pollination. Poor quality or insufficient pollen deposition can result in a poor fruit set and cause bunch failure, as observed in several studies [16,85,86,87,88]. The population density of *E. kamerunicus* can vary among the Dura x Pisifera (DxP) hybrid planting materials, which are correlated with different pollen quality, and the rate of female flower germination, both of which are crucial factors that affect seed and fruit production [89,90,91]. Low pollen viability can decrease fruit production in hybrids of *E. guineensis*, while high parthenocarpic fruit ratios are observed in the Oleifera x Guineensis (OxG) oil palm hybrid due to the bracteoles blocking the pollen’s entry, resulting in ineffective fertilization [89,91].

Pollen quantity and availability are also affected by the sex ratio of oil palm inflorescences, which can result in insufficient male inflorescences and negatively impact pollination efficiency [33,92]. Different oil palm varieties produce varying numbers and characteristics of male inflorescences as a source of pollen [90,91,93]. The sex ratio is generally determined by the genotypes and production of male inflorescences. Some varieties, such as DxP, Pisifera, and Tenera, are known to produce a high ratio of female to male inflorescences, which can cause a decrease in pollination due to limited pollen availability [33,93,94,95,96]. On the other hand, varieties such as ECPH550 produce a high quantity of male inflorescences, which leads to a higher weevil population. However, the ratio of female to male inflorescences can reach close to one as the oil palm matures [13]. When the palm ages, more male inflorescences are produced.

The inflorescence sex ratio refers to the proportion of female inflorescences produced by a palm tree. According to the Malaysian Palm Oil Board (MPOB) [97] and the International Society of Oil Palm Agronomists (ISOPA) [98], a healthy inflorescence sex ratio is typically between 65% and 70%. The ISOPA also states that a critical level for the inflorescence sex ratio is above 75%, which may result in a poor fruit set ratio. Conversely, a decrease in the inflorescence sex ratio below 75% to the healthy range could increase the fruit set ratio and fruit-to-bunch ratio. Conversely, an increase in the inflorescence sex ratio above 75% may lead to a decline in the fruit set ratio and fruit-to-bunch ratio.

To ensure successful pollination and reduce parthenocarpic fruitlets, a healthy inflorescence sex ratio must have enough male inflorescences to provide adequate pollen. Several factors can increase male inflorescence production, including planting material, frond production, soil type, and palm age. Research by Swaray et al. [5] revealed that both oil palm frond production and soil type play a significant role in male inflorescence production. Specifically, peat soil and tissue-cultured planting materials have been found to result in a decline in male inflorescence production.

Nevertheless, planting material is the most common external intervention used to alter the inflorescence sex ratio in oil palm. Breeding and selection in oil palm have historically focused on high oil yield and fresh fruit bunch production [99], resulting in the preference for female inflorescences that give rise to fruit-bearing palm bunches. This leads to a decrease in male inflorescence production, leading to a higher inflorescence sex ratio but lower fruit set, fresh fruit bunch yield, and oil yield. Research by Norman et al. [100] suggests that an increase in the number of female inflorescences may compromise male inflorescence production. Therefore, oil palm breeders should prioritize breeding and selecting high-quality planting materials that produce male flowers with viable pollen. Among oil palm DxP progenies, Deli Serdang × Cameroon has been reported to have the highest production of male inflorescences, with a mean value of 29.40% [5]. Another factor that can influence male inflorescence production in oil palm is aging, as reported by Corley and Gray [101], who found that the inflorescence sex ratio declines from 90% to below 60% as the palm ages when cultivated on Malaysia’s coastal soils, with cultivation ages of 4 and 15 years, respectively, after planting.

One study has shown that the population of *E. kamerunicus* increases as the oil palm ages, which is related to the number of spikelets in the oil palm inflorescences. The older the oil palm, the higher the spikelets, and thus the higher the *E. kamerunicus* population [57]. The compactness of the flower, represented by the number and length of the spikelets, affects the population abundance per bunch of *E. kamerunicus* [57]. The length of the anthesising spikelets is positively correlated with the number of *E. kamerunicus* on the male inflorescences [32]. Ref. [92] suggested that longer spikelets provide more breeding sites and food sources for the weevils. Insufficient male inflorescences may contribute to a decline in the *E. kamerunicus* population and reduce pollination activity, leading to poor fruit set production [89]. Pollination success is evidenced by fruit production.

However, data on the optimum number of available spikelets required to maintain the abundance of the *E. kamerunicus* population is still lacking. The optimum number of *E. kamerunicus* per spikelet is reported to be between 15 and 30 for good pollination, although varying reports exist [52,57]. These varying reports could be due to the genetic material of the oil palms, as well as soil types and palm age [57]. To ensure sufficient pollen supply, it is recommended to have at least two male palms per hectare in plantations with a high sex ratio, and assisted pollination can be used in plantations with insufficient male inflorescences [45,102].

## 4. Habitat Loss

Habitat loss is another major factor that can negatively impact pollinator populations. The conversion of natural habitats to agriculture, including oil palm plantations, reduces the availability of food, nesting sites, and shelter for pollinators, leading to declines in population size [103]. Various studies have looked at the impact of deforestation on *E. kamerunicus* and other oil palm pollinators. For example, a study by [104] found that deforestation led to a decrease in the diversity and abundance of oil palm pollinators, including *E. kamerunicus*, in Indonesia. Another study by [105] found that habitat destruction and fragmentation had a negative impact on the abundance of *E. kamerunicus* in Cameroon. Ref. [106] suggested that forested areas serve as important sources of food and habitat for the weevils.

Deforestation can cause a decline in the population of *E. kamerunicus* by reducing the availability of suitable habitats and food sources. Oil palm plantations that replace forests often lack the diversity of plant species that can provide alternative habitats and food sources for the weevil. This loss of habitat and food resources may decrease the population of *E. kamerunicus*, as well as other species that depend on these resources for survival. Additionally, deforestation can cause fragmentation of populations, isolating individuals and leading to decreased gene flow and reducing genetic diversity, which can further contribute to population declines and increase the risk of local extinction. Furthermore, deforestation can also lead to climate change, which can impact the availability of resources and the timing of flowering in oil palm trees. Changes in temperature and rainfall patterns can cause fluctuations in the timing of flowering, which can, in turn, affect the availability of pollen and nectar, as well as the timing of *E. kamerunicus* mating and reproduction cycles.

To address the issue of habitat loss that is causing the decline of pollinators, oil palm operators can take several specific conservation measures within their plantations. One crucial measure is retaining and restoring riparian vegetation, which can serve as essential habitat corridors for pollinators, allowing them to move freely across the landscape and access different resources. Additionally, establishing pollinator-friendly habitats, such as wildflower strips or bee hotels, can provide additional food and shelter for pollinators, encouraging their presence and activity within the plantation.

Another important conservation measure is reducing pesticide use and implementing integrated pest management practices. This involves minimizing the use of pesticides and adopting methods such as biological controls, which can help to minimize the negative impact of chemicals on pollinator populations, thus promoting their health and abundance. Monitoring and managing pollinator populations using regular surveys and data collection is also crucial for conservation efforts. By gathering data on pollinator populations, oil palm operators can make informed decisions and take appropriate actions to protect and support pollinators within their plantations.

Furthermore, educating workers and local communities about the importance of pollinators and their role in oil palm cultivation can foster a culture of conservation. This can include training programs, workshops, and awareness campaigns to raise awareness about the need to protect pollinators and their habitats. Collaborating with stakeholders, such as researchers, NGOs, and local communities, is also vital in protecting pollinator populations. By working together, oil palm operators can promote collective action toward pollinator conservation, including partnerships for research, conservation initiatives, and community engagement programs.

Finally, adhering to recognized sustainability standards, such as RSPO (Roundtable on Sustainable Palm Oil) certification, can ensure that oil palm operations are environmentally responsible and socially inclusive. By following these standards, oil palm operators can demonstrate their commitment to pollinator conservation and promote sustainable oil palm cultivation practices. In summary, by implementing a combination of measures such as retaining riparian vegetation, establishing pollinator-friendly habitats, reducing pesticide use, monitoring and managing pollinator populations, educating workers and local communities, collaborating with stakeholders, and adhering to sustainability standards, oil palm operators can contribute to the protection and conservation of pollinators, promoting sustainable oil palm cultivation practices that are beneficial for both pollinators and the environment.

## 5. Proximity of Oil Palms to Natural Forests

The expansion of the oil palm industry has led to the cultivation of marginal land, resulting in a substantial increase in the size of plantations, particularly in Southeast Asia. This expansion has been a major contributor to mass deforestation in the region [106] and the displacement of other agricultural land use, systems such as rubber agroforestry. Malaysia alone had 5.87 million hectares of oil palm planted area in 2020 [107]. Large-scale oil palm agriculture typically takes place in diverse natural rainforest areas, resulting in isolated forest patches being surrounded by oil palm monocultures. Environmental non-governmental organizations (NGOs) have raised concerns and criticism over the global conversion of large natural tropical rainforests for oil palm monoculture. In response, the Roundtable on Sustainable Palm Oil (RSPO) was established in 2004 to improve the environmental performance of both producers and users [108]. One of the implemented criteria is to retain areas that support high conservation values (HCVs) in the plantation [106,109]. These areas need to be identified and maintained to preserve their role as habitats for diverse flora and fauna and important ecosystem services. Therefore, it has become mandatory to preserve adjacent natural forests to comply with and support sustainable oil palm agroforestry.

Higher proportions of natural environments in a landscape are also important for maintaining pollinator populations [110,111]. However, the relationship between the visitation rates and diversity of insect pollinators with the presence of natural forests within a certain distance varies depending on factors such as the plantation site, type of natural forest, and insect pollinators. The spatial organization of a landscape greatly impacts the survival and dispersal ability of most pollinator species, as resource availabilities are affected by spatial establishment and functional connectivity [112,113]. The spatial pattern of species, including insect pollinators, is influenced by biotic interactions such as competition, predation, and parasitism [10,12,40,114]. The efficiency of animal-mediated pollination services is significantly impacted by biotic interactions, such as competition between pollinators [80,115]. For instance, *E. kamerunicus* is also influenced by these biotic interactions, including predatory threats and parasitism [12,40].

## 6. Climate and Weather Driver

Rainfall is one of the most studied climatic factors and is often linked to insect activities, particularly pollination. It is reported to affect pollination efficiency since pollen in wet flowers is hard to disperse [17]. The activity of the *E. kamerunicus*, in particular, is negatively correlated with rainfall [34,52,77]. Heavy rains can decrease pollination efficiency because they reduce inflorescence visits. While there may be a low rate of pollen transfer, pollen can accidentally be removed from the weevils’ bodies [27,57,76,116]. A reduction in pollen densities carried on the weevil’s body would mean an inadequate pollen supply, resulting in a low oil palm fruit set [57,76,117]. Additionally, wet weather may favor parasitic nematodes, as reported in Southeast Asia [27]. Although the weather can affect pollination efficiency, the claim is still inconclusive and requires further research.

The impacts of high rainfall on the *E. kamerunicus* are also documented to be related to the distribution of rainfall, as well as the amount and the number of rainy days. Ref. [118] reported that the distribution of rainfall even a month prior to the data collection impacts the flight activity of *E. kamerunicus* on male inflorescences, especially with rain levels exceeding 500 mm. Changes in the number of rainy days can also alter the pollinating activity of *E. kamerunicus* since it reduces the pollen load and its viability [92]. Furthermore, pollen viability is also affected during rainy days. During the dry season, pollen is less humid, and its germination capacity is preserved [27,116]. Pollen production is also reduced during the rainy season due to high humidity [13]. The dry season is thus preferable for the pollinating roles of the *E. kamerunicus*, as it provides a better environment for pollen transfer and pollen quality. Pollinating efficiency was also observed to be maintained during the dry season with an average rainfall of less than 200 mm/month, even when the weevil population declined [13,57].

Nevertheless, a few other observations documented otherwise, whereby there is a declining pattern in the *E. kamerunicus* population with a lack of rain [34,52,92]. The weevil population increased linearly with higher rainfall distribution, the number of rainy days, and humidity [12]. The population development of *E. kamerunicus* was reported to be faster during the rainy season, thus suggesting that it could be combined with other insect pollinators to maintain pollination activity during the rainy season [34,40,52].

Fluctuations in other climatic factors also impact the weevil and its pollination service. In addition to rainfall, temperature is also noted to affect the flight activity and population density of *E. kamerunicus* [118]. A rise in temperature causes a decrease in the weevil’s abundance, which may negatively affect fruit production. Ref. [115] claimed that the impact of changing temperature is greater than the fluctuations of rainfall. High wind velocity is another factor that is responsible for the variations in the *E. kamerunicus* population [54]. In tall palms, strong winds at the canopy level can limit the weevil population, leading to lower numbers.

The growth of oil palm inflorescences and spikelets is also influenced by multiple climatic factors [89]. For example, ref. [27] found that when underwater and affected by radiation stress, more male inflorescences are produced. Similarly, ref. [5] observed that water availability has a significant effect on oil palm growth. Changes in climatic conditions, such as temperature, precipitation, and vegetation cover, also have an impact on the weevils’ population dynamics [52,118]. These changes are often influenced by landscape features, including altitude and topography [54]. Over time, variations in climatic patterns can significantly affect plant–pollinator interactions, including pollination activity [116].

## 7. Conclusions and Recommendations

The cultivation of oil palms is a topic of great debate due to its significant commercial value and notable impacts on the ecosystem. There are numerous issues related to palm oil production that have prompted calls to boycott products containing this ingredient. However, even the most fervent environmentalists agree that boycotting palm oil is not the solution. Therefore, there is a need for intensive research to improve the sustainability of the industry and ensure benefits for people, the planet, and profitability.

In Malaysia, a plantation area cap has been set to limit the total oil palm cultivated area to 6.5 million hectares. To increase productivity, the country must focus on improving technology and productivity per hectare, particularly on improving fruit set formation. This has prompted research into the variables that could affect pollination efficiency in oil palms.

Several determinants influence oil palm pollination, which can be divided into controllable and uncontrollable factors (Figure 1). Controllable factors include the size and complexity of the pollinators’ population, pollen-carrying capacity, oil palm varieties, the number of male inflorescences, and the biodiversity of oil palm estates. Uncontrollable factors are mainly climate drivers such as air temperature, atmospheric pressure, humidity, precipitation, solar radiation, and wind. Despite intensive research on oil palm pollination, several research gaps still exist. A focus on the impacts of other climate drivers, such as solar radiation and drastic weather on oil palm pollination and fruit formation, is imperative. On top of that, further studies on how abiotic (e.g., the foraging behavior of multi-pollinator communities) and biotic factors (e.g., the effects of inflorescence spatiotemporal sex distribution on pollinators) contribute to the pollination efficiency in oil palm are also crucial. A better understanding of the pollinator community and the role of biodiversity in this agroecosystem can contribute to the crop’s environmental and economic sustainability.

## Figures and Tables

**Figure 1 insects-14-00454-f001:**
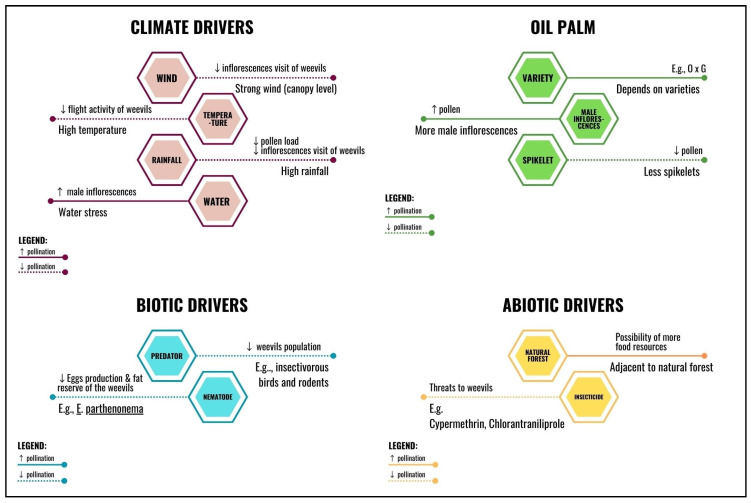
Common factors affecting pollination activity in the oil palm plantation.

**Table 1 insects-14-00454-t001:** Pollination techniques for enhancing pollination in oil palm plantations.

Technique	Methods	Pollination Efficiency	Limitations
Hatch and Carry application [11,12,13]	i. In each box, six male inflorescences (4 to 5 days post-anthesis) are put inside. Inflorescences are replaced every 9 to 12 days. In order to check any hatched weevils, five male inflorescences are removed daily.	i. An increase of between 15.04% and 21.05% in the fruit set after weevil release. The ideal distance between the hatch and carry boxes is 400 m or about one to two boxes per block of about 25 ha.	i. Observations made 4 to 6 months after release indicated a low increase in the weevil population, which may be attributed to limited breeding sites. This is likely due to a relatively small increase in the number of anthesizing male inflorescences, which only rose 1.5 times compared to the pre-observation period.
	ii. Before release, the weevils are sprayed with 1 g of high-viability pure pollen. Fruit set values of oil palms at five different distances from each box were analyzed (10 m, 100 m, 200 m, 300 m, 400 m).	ii. *E. kamerunicus* population increased by 32 to 45% after this technique was applied.	
Hand pollination [14]	i. Pollen was collected from anthesizing male inflorescences of commercial Dura x Pisifera (DxP) palms and was incubated in a 10% sucrose solution containing 15 drops of 5% boric acid solution to test its viability. Only pollen with more than 60% viability was used for hand pollination. Pollen was mixed with 2 g of talcum powder before puffing on the anthesizing female inflorescences in one application.		i. The fruit set was lower compared to open pollination, which could be due to the limited number of receptive flowers during hand pollination. Additionally, hand-applied pollen might be difficult to reach the inner flowers. Weevils were more efficient pollinators since they forage in the inflorescences, while pollen was applied only once in hand pollination.
	ii. The amount of pollen used was 0.0001. 0.001, 0.01, 0.1, 1.0, and 5.0 g per anthesizing inflorescence. The young inflorescences were bagged at least a week before anthesis, and hand pollinated at the first sign of anthesis by injecting pollen through a small hole in the bag, which was then re-closed.		ii. Higher parthenocarpic fruits in the inner bunch with the fruit set.
Wind- and weevil-assisted pollination [15]	Wind pollination—inflorescences were covered with fine nylon bags to prevent the entry of the pollinating weevils. Weevil pollination—inflorescences were covered with transparent plastic bags with openings at the bottom to facilitate the entry of the weevils. Wind + Weevil pollination—inflorescences were left open where the pollination was affected by both wind and weevils. Wind + Assisted pollination—inflorescences were covered with nylon-mesh bags and were given three rounds of assisted pollination from the first to the third day of anthesis. Wind + Weevil + Assisted pollination—inflorescences were left open to facilitate pollination by wind and weevils and were supplemented with two rounds of assisted pollination.	i. Wind + Weevil + Assisted pollination resulted in the highest fruit set. ii. Wind-pollinated bunches recorded the highest parthenocarpic fruit: weight ratio. iii. Weevil pollination decreased the parthenocarpy from 34.3% to 24.3%. Though wind + assisted pollination was more efficient than wind + weevil pollination, the latter produced compact bunches owing to the pollination of even the spikelets at the bottom of the inflorescences by weevils.	
Complementing-assisted pollination with artificial pollination [16]	Assisted pollination was carried out during anthesis by applying the pollen to the receptive female inflorescences. Subsequently, on the 7th and 14th days after anthesis, 1-naphthaleneacetic acid (NAA) was applied. In total, there were three applications for the inflorescences.	i. Artificial pollination with the application of naphthaleneacetic acid (NAA) is an economically viable method for increasing the productivity and profitability of the hybrid Oleifera x Guineensis (OxG). Artificial pollination shows clear benefits for FFB producers.	i. Additional costs in inputs and labor. ii. An increase in the cost of pollination, a larger amount of fertilizer, and transportation costs.
		ii. An increase in the production of FFB by 12.4%, and an increase in net income by 7.7% per hectare of crop.	

**Table 2 insects-14-00454-t002:** Key characteristics of *E. kamerunicus* as the main pollinator of oil palm in Southeast Asia.

Characteristics
Dependence on the male inflorescence as their single host. Large pollen-carrying capacity. Ability to penetrate deeper into male inflorescences. Ability to co-exist with other native pollinators. Collect only fresh and viable pollen.

**Table 3 insects-14-00454-t003:** Complementarity behavior of *E. kamerunicus*, *Thrips hawaiiensis,* and *Pyroderces* sp. in the oil palm plantation.

Characteristics	*Elaeidobius kamerunicus*	*Thrips hawaiiensis*	*Pyroderces* sp.
Active time	Morning to mid-day ^a^	Early morning to late afternoon	Sunset ^d^
Pupation habitats	Male inflorescence ^d^	Soil ^c^	Male inflorescence ^d^
Most efficient period	Wet ^d^	Dry ^d^	ND
Activity in the female inflorescences	Feeding and ovipositing ^c^	Feeding ^c^	Ovipositing ^c^
Activity in the male inflorescences	Incidental ^c^	Inhabits ^b^	Visits ^c^

Note: ND—no data; source: ^a^ [31]; ^b^ [19]; ^c^ [27]; and ^d^ [34].

## Data Availability

Not applicable.

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
