# Peer review of "Factors Affecting Pollination and Pollinators in Oil Palm Plantations: A Review with an Emphasis on the Elaeidobius kamerunicus Weevil (Coleoptera: Curculionidae)"

_insects, 2023, doi:10.3390/insects14050454_

Round 1

Reviewer 1 Report

Referee report for Insects – March 2023

Ref:  [Insects] Manuscript ID: insects-2304533 entitled « Factors affecting pollination and pollinators in oil palm plantations: A review with Emphasis on the Elaeidobius kamerunicus weevil”

by Christharina S Gintoron , Mohamad Azmi Mohammed , Elvy Quatrin Deka , Siti Nurlydia Sazali , Kian Huat Ong , Imran Haider Shamsi, Patricia Jie Hung King

General comments

The primary aim was to propose a review on factors affecting pollination and pollinators Elaeidobius kamerunicus in oil palm plantations

This paper is reviewed whereas another review has been recently published in March 2023 in Journal of Oil Palm research 35 (1) entitled “Impact of Elaeidobius kamerunicus (Faust) introduction on oil palm fruit formation in Malaysia and factors affecting its pollination efficiency: A review” by  Saharul Abillah Mohamad et al.

During the last few years, a certain number of reviews have been published such as :

·         Appiah and Agyei-Dwarko 2013

·         Li et al. 2019,

·         Zulkefli et al. 2020

·         Mamehgol Yousefi et al. 2020 [99]

Nevertheless, this review is interesting because pollination is an important topic for palm fruit formation and yield.

What is boring while reading is the confusion between the continents where the situation is completely different.

Also, the references using figures [XX] is not easy at all for the reader. It is difficult to refer to the list of references all the time. 

The authors pointed out many important and recent references.

The different paragraph titles are often inappropriate to the content of the paragraphs

From my reading, the review appears to have been conducted rigorously but there are a lot of inaccuracies. 

Generally, the paper is clearly written.

More significant issues

Abstract

▪ Line 33: “the primary pollinator of oil palm male inflorescences”. It is confuse because in the abstract we do not know which continent is concerned: South East Asia ? where only Elaeidobius kamerunicus has been introduced or Africa ? where Elaeidobius kamerunicus is originated from but is not the only Elaeidobius species and is not “primary”.

1.         Introduction:

▪ Line 70: I do not agrre with the word “to protect pollinator pollination”.

▪ Line 73:  Add “Elaeis guineensis Jacquin

▪ Line 79:  Be careful when writing “female inflorescences” and “female flowers”. It might be better to describe before line 74.

“The female inflorescences are found on the axillary branches”

▪ Line 84:  What about anemophily pollination? It has to be mention at the stage too as entomophil pollination.

▪ Line 96:  Table 1 is an interesting contribution to the review

2.       Oil Palm Pollination agents           

▪ Line 107:  “a greater number of entomofauna”. It would be important to know which continent we are talking about. We need to go through the references [20, 23] to understand that is the cas in Africa.

▪ Line 109-110:  Very important errors: Oryctes rhinoceros is a pest affecting the crown and not at all the inflorescences (even in rare cases it may attack the pedoncule of the inflorescence). In addition, it is not a Curculionid beetle, O. rhinoceros is a Coleoptera Scarabaeidae. O. rhinoceros is a pest of oil palm in South East Asia. So, again confusion regarding the continent we are talking about.

In the same way, Rhynchophorus palmarum is a pest attacking the crown, not the inflorescences.  It is not a chrysomelid beetle but a Coleoptera Curculionidae. I specify that R. palmarum is affecting palms in South America.

The mention of these two pests does not have to be mention in the review.

So, a lot of confusion between continents for the reader.

▪ Line 115: Again all different species of Elaeidobius mentioned between Lines 111 to 115 are found in Africa, not elsewhere. This must be clearly stated. The reference Desmier de Chenon 1981 has to be mentioned Line 114. Entomophile pollination of oil palm in West Africa. Preliminary research. In: The oil palm in agriculture in the eighties. Incorporated Society of Planters ed. Malaysia 1: 239-291.

▪ Line 126:  about the volatile compounds, it needs a reference there about the different compounds mentioned

▪ Line 138: Pheromone is not the appropriate word there. A pheromone is a chemical signal, which attract one sex to another sex, and is specific to the insect species. It has nothing to do with odors emitted by a plant. It plays in the communication between individuals of the same species. Insects, not plant, produce pheromone.

▪ Line 139-140: The pheromone is not produce by oil palm. Please, replace pheromone by volatile compounds for example.

▪ Line 148:  Please check with the contains of the thesis ref [18] Auffray et al. 2017.

▪ Line 149:   This paragraph has to move before .. Lines 122…..

▪ Line 153:   Need a reference “only offer nectar”

▪ Line 155:   a space to be removed before “Moreover”

▪ Line 155-156:   More flower blooms earlier”? I am not aware of that. Which reference?

Important

The content does not correspond to the title of the paragraph entiled Elaeidobius kamerunicus. The authors refer a lot to the volatile compounds and the difference between male and female inflorescences compounds emitted. This is not mentioned in the title of 2.1. Please, change the title of 2.1.

▪ Line 169:    The space to be removed between E. and kamerunicus.

▪ From Lines 120 to 178, there are many repetitions

▪ Line 174:  Weevils are Curculionidae. Therefore, Weevils is enough.

▪ Line 180:  Please, specify that this is the Asian continent “Elaeidobius kamerunicus is the main pollinator…” because it has been introduced in the 1980’.

▪ Line 189:  Figure 1 is not clear; a table would suffice to give the characteristics.

▪ Line 182: There the authors should mention mutualistic relation between Elaeidobius sp and oil palm.  Oil palm E. guineensis is host specific for E. kamerunicus but please check for E. oleifera for which E. kamerunicus is not host-specific (refers to [18]). E. kameruncius may be host by the hybrid but not E. oleifera alone.

▪ Lines 183-184: Repetition with 2.1 again

▪ Lines 186-188. Which reference(s)?  Which country? It depends of the seasons. It is not true at all.

▪ Lines 192 and 195: Repetition.

▪ Lines 191-203: This morphology of E. kamerunicus is a part which would be more suitable in the 2.1 entitled Elaedobius kamerunicus.

▪ Lines 204-205: Repetition again.

▪ Lines 211-214: This biology. It might be mentioned before in 2.1

▪ Line 214: Nothing about data regarding life cycle?

▪ Line 215: Repetition with Line 206. This specific and mutualistic relation has already be mentioned before.

▪ Line 218: Again repetition with Line 210.

▪ Line 221: It is only here the authors mentionned “Mutualistic relation”. It has to be written before and then explaining what means mutualism.

▪ Lines 225-233: This paragraph compares hand pollination and natural pollination.

▪ Line 241: Above (Line 237) is mentioned “other native pollinators” and reference to Thrips and Pyroderces but what about the other pollinators in the native area in West Africa where other Elaeidobius species. Do these have different host plants ?

▪ Lines 248-250: Table 2. Is it sure that the wet conditions are the most efficient period for E. kamerunicus. It is certain that the insect is better adapted to rainy condition than the other Elaeidobius species (see lines 255-257).

In the Characteristics column, there are 2 lines “Activity in the male inflorescences”?

▪ Line 262:  Delete the sentence which is repeated  lines 263-264.

▪ Line 279: “Female” inflorescences do not produce pollen !!!!!!

Moreover, E. kamerunicus need to complete its life cycle during male inflorescence anthesis so, probably it is attracted to male inflorescences at the first days of anthesis.

▪ Line 289: It needs italic for E. kamerunicus

▪ Lines 291-292: Already mentioned. Repetition

▪ Lines 296- 307: What about the number of male inflorescences per hectare ? Whate about the planting material origin? This paragraph is quite poor…

▪ Line 311. Why “including E. kamerunicus”. It is the title of the article, is not it ? Sentence to rephrase, please.

▪ Line 345: Reference to Syed?

▪ Line 354: Remove a space before [72]

▪ Line 368: Repetitions “such as isolation, limited dispersal” (see Line 351)

▪ Line 380: Cosmocercoides sp. in italic

▪ Line 382: “In one study” Repetition. Please, reformulate the sentence

▪ Lines 384-385: A nematode is an entomopathogen, it cannot competed for nutrients with Elaeidobius? Nutrients for nematode are Elaeidobius, live individuals.

▪ Lines 394-395: This sentence has to be moved.

▪ Lines 404-405: “as well as “ is repeated twice.

▪ Line 407: “and may have “ to be reformulated

3.        Inadequate pollen deposition and pollen quality

▪Line 454: Reference is needed

▪ Lines 459-460: “attracting more weevils”. Not necessarly if the population of weevils is low

▪ Line 463: “making them more attractive”.  No. Not demonstrated. Reference ?

4.       Habitat loss

5.       Impact of proximity of palms to natural forests on Elaeidobius kamerunicus

6.       Climate and weather driver

Rainy season, dry season, temperature, wind and other factors are explored in this paragraph.

▪ Lines 554-558: Repetition. To be reformulated.

7.       Conclusion and recommendation

▪ Lines 577 to 583: To be reduced  to 3 lines. Too long

▪ Lines 586 to 590: Too long

▪ Lines 604: Table 3 must be completely revised. It needs major revisions. It may be choose to include these future researches in the text himself.

References

▪ Line 631:  [8] et al. ? to be completed with the co-authors

▪ Line 648:  [16] et al. ? to be completed with the co-authors

▪ Line 649:  [16]  Elaeis guineensis (italic)

▪ Line 655:  [19] et al. ? to be completed with the co-authors

▪ Line 664:  [23] The authors are N. Hala, Y. Tuo, A.A.M. Koua and Y. Tano.

▪ Line 666:  [24] et al. ? to be completed with the co-authors

▪ Line 671:  [26] Thrips hawaiiensis in italic

▪ Line 676:  [28] et al. ? to be completed with the co-authors

▪ Line 683:  [31] et al. ? to be completed with the co-authors

▪ Lines 690-693:  [34] doi only

▪ Line 698: [36] Dimorphim

▪ Line 704:  [39] Please complete “et al.” by the co-authors.

▪ Lines 706-710: [40] Need revision

▪ Line 717: [44] Number of pages is missing

▪ Lines 719-723: [45] Need revision

▪ Line 748: Faust

▪ Lines 764-765:  revision

▪ Line 770 [65]: et al. ? to be completed with the co-authors

▪ Line 779 [69]: et al. ? to be completed with the co-authors

▪ Line 782 [70]: Elaeis guineensis in italic

▪ Lines 791-795:  revision

▪ Lines 802-806:  revision

▪ Lines 809-814 [80]: et al. ? to be completed with the co-authors

▪ Line 815 [81]: et al. ? to be completed with the co-authors

▪ Line 853: In International

▪ Line 855 [98]: et al. ? to be completed with the co-authors

▪ Line 861 [100]: et al. ? to be completed with the co-authors

▪ Line 863 [101]: et al. ? to be completed with the co-authors

▪ Line 868 [103]: et al. ? to be completed with the co-authors

▪ Line 871 [105]: et al. ? to be completed with the co-authors

▪ Line 873 [106]: et al. ? to be completed with the co-authors

▪ Line 875 [107]: et al. ? to be completed with the co-authors

▪ Line 880 [109]: et al. ? to be completed with the co-authors

▪ Line 882 [110]: et al. ? to be completed with the co-authors

▪ Line 885 [111]: revision

▪ Line 890 [113]: et al. ? to be completed with the co-authors

Conclusion

This information is of interest to agronomists, botanists and entomologists working with oil palm and merits publication.

Based on the comments above reported, my opinion is that this manuscript may be suitable for printing on this journal BUT after major revisions.

Reviewer 2 Report

Article is worth reading. Still chances are existing for inclusion of some other information related to male flower initiation using external treatments in problematic areas and high sex ratio palms. Please refer ISOPA2019.

Reviewer 3 Report

After carefully reviewing the manuscript, I believe that your manuscript has the potential to make a valuable contribution to the literature on pollination in oil palm plantations. With some revisions and improvements, it could be a strong addition to the field. I have attached my comments for improvement

Round 2

Reviewer 1 Report

My comments and change requirements are mentionned in the file itself
